

# Validation of meteorological analyses and trajectories in the Antarctic lower stratosphere using Concordiasi superpressure balloon observations

Lars Hoffmann[1], Albert Hertzog[2], Thomas Rößler[1], Olaf Stein[1], and Xue Wu[1,3]

[1]Jülich Supercomputing Centre, Forschungszentrum Jülich, Jülich, Germany
[2]Laboratoire de Météorologie Dynamique, École Polytechnique, IPSL, Palaiseau, France
[3]Institute of Atmospheric Physics, Chinese Academy of Sciences, Beijing, China

*Correspondence to:* L. Hoffmann (l.hoffmann@fz-juelich.de)

**Abstract.** In this study we validated temperatures and horizontal winds of meteorological analyses in the Antarctic lower stratosphere, a region of the atmosphere that is of major interest regarding chemistry and dynamics of the polar vortex. The validation was performed with long-duration observations from 19 superpressure balloon flights during the Concordiasi field campaign in September 2010 to January 2011. Our intercomparison covers the European Centre for Medium-Range Weather Forecasts (ECMWF) operational analysis, the ERA-Interim reanalysis, the Modern-Era Retrospective analysis for Research and Applications (MERRA), and the National Centers for Environmental Prediction and National Center for Atmospheric Research (NCEP/NCAR) reanalysis. We found that large-scale temperatures of the analyses have a mean precision of $0.4 - 1.4$ K and a warm bias of $0.4 - 2.1$ K at about $17 - 18.5$ km altitude and $60 - 85°$S. Zonal and meridional winds have a mean precision of $0.9 - 2.3$ m s$^{-1}$ and a bias below $\pm 0.5$ m s$^{-1}$ in the same region. Standard deviations related to small-scale fluctuations such as gravity waves are reproduced at levels of $15 - 60\%$ for temperature and $30 - 60\%$ for the horizontal winds. We also used the balloon observations to validate trajectory calculations, where vertical motions of simulated trajectories were nudged to pressure measurements of the balloons to take into account changes in the overall mass configuration of the balloon-gondola system. We found relative horizontal transport deviations of $4.5 - 12\%$ and error growth rates of $60 - 170$ km day$^{-1}$ for 15-day trajectories. Dispersion simulations revealed some difficulties with the representation of subgrid-scale wind fluctuations in our Lagrangian transport model, as the spread of air parcels simulated with different analyses was not consistent. Although case studies suggest that the accuracy of trajectory calculations is influenced by meteorological complexity, diffusion generally does not contribute significantly to transport deviations in our analysis. Overall, validation results are satisfactory and compare well to earlier studies using superpressure balloon observations. In most cases, best performance was achieved by the ECMWF operational analysis, having the best spatiotemporal resolution, followed by ERA-Interim, MERRA, and finally NCEP/NCAR, having the lowest spatiotemporal resolution. Future work applying Eulerian or Lagrangian models to study the chemistry and dynamics of the polar vortex may use our validation results as additional guideline for error analyses.



# 1 Introduction

The seasonal formation and decay of the polar vortex is likely the most prominent feature of the extratropical stratospheric circulation (e. g., Schoeberl and Hartmann, 1991; Newman and Schoeberl, 2003; Waugh and Polvani, 2010). The structure and dynamics of the polar vortex play a key role in the winter and spring stratospheric circulation and coupling between the

stratosphere and troposphere. A number of studies demonstrated that the polar vortex can influence tropospheric weather and climate (Baldwin and Dunkerton, 1999; Polvani and Kushner, 2002; Thompson et al., 2002; Baldwin et al., 2003). Furthermore, the polar vortex acts as a cold trap for stratospheric air, which plays a critical role in polar ozone depletion and the annual formation of the Antarctic ozone hole (Solomon, 1999, and references therein). These topics motivated various observational and modeling studies in recent years to better understand the structure and dynamics of the polar vortex as well as implications

on polar ozone loss in the stratosphere.

Lagrangian particle dispersion models are indispensable tools to study atmospheric transport processes (e. g., Lin et al., 2012). Trajectory calculations in Lagrangian transport simulations are commonly driven by wind fields from global meteorological reanalyses. The accuracy of trajectory calculations depends on various factors, including interpolation and sampling errors related to the finite spatial resolution of the meteorological data as well as errors of the wind field itself, which are

introduced during the data assimilation process (e. g., Stohl, 1998; Bowman et al., 2013). In this study we aimed at direct validation of temperature and wind data as well as trajectory calculations for the Antarctic lower stratosphere using different meteorological data sets. We assessed the performance of three meteorological reanalyses, including the European Centre for Medium-Range Weather Forecasts (ECMWF) ERA-Interim reanalysis (Dee et al., 2011), the Modern-Era Retrospective analysis for Research and Applications (MERRA) reanalysis (Rienecker et al., 2011), and the National Centers for Environmental

Prediction and the National Center for Atmospheric Research (NCEP/NCAR) reanalysis (Kalnay et al., 1996). Furthermore, we compare with the ECMWF operational analysis (OA), which is produced with significantly higher spatial resolution.

For validation we utilized superpressure balloon observations during the Concordiasi field campaign (Rabier et al., 2010) in September 2010 to January 2011. During the campaign 19 superpressure balloons were launched from McMurdo station (78°S, 166°E), Antarctica. Each balloon flew in the mid- and high-latitude lower stratosphere for a typical period of three months.

The sensors aboard the balloons provide position, pressure, and temperature at high accuracy and high temporal sampling. Various studies demonstrated that superpressure balloon observations constitute an excellent source of data for the validation of meteorological analyses (Knudsen et al., 1996, 2002; Hertzog et al., 2004, 2006; Knudsen et al., 2006; Boccara et al., 2008; Podglajen et al., 2014; Friedrich et al., 2017). This paper presents an update on earlier work, in particular to Boccara et al. (2008), who performed a validation analysis based on superpressure balloon observations during the Vorcore campaign in

Antarctica in September 2005 to February 2006. The results are also compared with findings of the PreConcordiasi campaign (Podglajen et al., 2014), which took place at tropical latitudes in February 2010.

Our new study may contribute directly to current research activities that focus on intercomparisons of different reanalyses, e. g., the Stratosphere–troposphere Processes And their Role in Climate (SPARC) Reanalysis Intercomparison Project (S-RIP) (Fujiwara et al., 2016). Here we applied the Lagrangian particle dispersion model Massive-Parallel Trajectory Calculations



(MPTRAC) (Hoffmann et al., 2016a) to conduct the trajectory calculations for the balloon observations. MPTRAC is a rather new model and our study also serves the purpose of validating the model. However, the results are transferable to other Lagrangian models for the stratosphere as well, e. g., the Chemical Lagrangian Model of the Stratosphere (CLaMS) (McKenna et al., 2002a, b) or the Alfred Wegener Institute Lagrangian Chemistry/Transport System (ATLAS) (Wohltmann and Rex, 2009). The results of the direct validation of the temperature and wind data of the meteorological analyses are of interest for studies using chemistry-transport models to assess polar ozone loss in the stratosphere (e. g., Chipperfield, 1999; Grooß et al., 2002, 2005; Wohltmann et al., 2013). The results of the trajectory validation are of particular interest for studies applying the 'Match' technique (von der Gathen et al., 1995; Rex et al., 1997) to assess polar ozone loss. In order to distinguish between chemically and transport-induced changes of ozone abundance, the Match approach uses trajectory calculations to relate ozone observations within the same air mass at different locations to each other.

In Sect. 2 we introduce the superpressure balloon observations during the Concordiasi campaign. We also describe the four meteorological data sets and discuss the meteorological conditions during the campaign. Furthermore, we introduce the Lagrangian particle dispersion model MPTRAC and the approach used for trajectory validation. The results of our study are provided in Sect. 3. In the first part we compare temperatures and horizontal winds of the different meteorological data sets directly at the position of the balloon measurements. In the second part we focus on the validation of trajectory calculations. Finally, Sect. 4 provides a summary and conclusions.

## 2   Data and methods

### 2.1   Superpressure balloon observations

Superpressure balloons are aerostatic balloons, which are filled with a fixed amount of lifting gas, and where the maximum volume of the balloon is kept constant by means of a closed, inextensible, spherical envelope. After launch, the balloons ascend and expand until they reach a float level where the atmospheric density matches the balloon density. On this isopycnic surface a balloon is free to float horizontally with the motion of the wind. Hence, superpressure balloons behave as quasi-Lagrangian tracers in the atmosphere. In this study we analyzed superpressure balloon observations in the lower stratosphere during the Concordiasi field campaign in Antarctica in September 2010 to January 2011. The Concordiasi field campaign aimed at making innovative atmospheric observations to study the circulation and chemical species in the polar lower stratosphere and to reduce uncertainties in diverse fields in Antarctic science (Rabier et al., 2010). During the field campaign 19 superpressure balloons with 12 m diameter were launched from McMurdo station (78°S, 166°E), Antarctica by the French space agency, Centre National d'Etudes Spatiales (CNES). Balloons of this size typically drift at pressure levels of ~60 hPa and altitudes of ~18 km. The balloons were launched between 8 September and 26 October 2010, and each balloon flew in the mid- and high-latitude lower stratosphere for a typical period of two to three months. The flight dates are summarized in Table 1 and the balloon trajectories are shown in Fig. 1.

The positions of the balloons were tracked over time by means of global positioning satellite (GPS) receivers. At each observation time the components of the horizontal wind are computed by finite differences between the GPS positions. The





uncertainty is about 1 m for the GPS horizontal position and $0.1\,\mathrm{m\,s^{-1}}$ for the derived winds (Podglajen et al., 2014). Each balloon launched during Concordiasi was equipped with a meteorological payload called Thermodynamical sensor (TSEN). TSEN makes in-situ measurements of atmospheric pressure and temperature every 30 s during the whole flight. The pressure is measured with an accuracy of 1 Pa and a precision of 0.1 Pa. The air temperature is measured via two thermistors. During

daytime, the thermistors are heated by the sun, leading to daytime temperature measurements warmer than the real air temperature. An empirical correction has been used to correct for this effect (Hertzog et al., 2004). The precision of temperature observations is $\sim 0.25\,\mathrm{K}$ during daytime and $\sim 0.1\,\mathrm{K}$ during nighttime. Note that technical issues aboard the scientific gondola caused a few data gaps in the TSEN data set, but most of them were shorter than 15 min.

In order to quantify the coverage of the balloon observations during the free-flying phases, we independently calculated the

5% and 95% quantiles of various parameter distributions. All statistics presented in this paper are most representative for the parameter ranges reported below. Any findings for parameters outside these ranges need to be considered carefully, because only few measurements are available to support them. We found that most of the measurements (i. e., more than 90%) took place between 25 September and 22 December 2010, at an altitude range of $17.0 - 18.5\,\mathrm{km}$, and within a latitude range of $59 - 84°\mathrm{S}$. The pressure measurements are mostly within a range of $58.2 - 69.1\,\mathrm{hPa}$ and the temperature measurements within

$189 - 227\,\mathrm{K}$. The density of air, calculated from pressure and temperature, varies between $0.099 - 0.120\,\mathrm{kg\,m^{-3}}$. The zonal winds are predominately westerly and mostly within a range of $1 - 44\,\mathrm{m\,s^{-1}}$. The meridional wind distributions are nearly symmetric, with meridional winds being in the range of $\pm 17\,\mathrm{m\,s^{-1}}$. Horizontal wind speeds are mostly within $5 - 47\,\mathrm{m\,s^{-1}}$.

As an example, Fig. 2 shows time series of density, temperature, zonal wind, and meridional wind as measured during flight number 4 of the Concordiasi campaign. The density time series shows decreasing density during the first 20 days, but remains

rather stable thereafter. This initial decrease in density is due to the release of dropsondes, which are another part of the balloon payloads on flight number $1 - 13$. The release of dropsondes changes the overall mass configuration of the balloon-gondola system, which is compensated by changes in density. A closer inspection of the time series reveals also diurnal variations in the balloon density. During the day the balloon envelop is heated by the sun, which increases the temperature and pressure of the gas inside the balloon. The balloon slightly expands in return, which decreases its equilibrium density. In addition to this

regular daily pattern, the time series show notable variability on even shorter time scales, including semi-diurnal oscillations of the horizontal winds, which are attributed to near-inertial gravity waves and semi-diurnal tides. As we do not expect the reanalyses to reproduce those fluctuations with great accuracy, we applied a low-pass filter with 15 h cut-off period to suppress all oscillations caused by pure and inertia-gravity waves, so that only perturbations due to large-scale dynamics remain visible in the time series (cf. Fig. 2). The cut-off period of the low-pass filter was selected to cover the longest inertial period in the

balloon data set, $T = 2\pi f^{-1}$, with Coriolis parameter $f$, ranging from about 12.0 h at 85°S to 13.9 h at 60°S.

## 2.2 Meteorological data

In this study we considered four meteorological data sets, the ECMWF operational analysis, ERA-Interim (Dee et al., 2011), MERRA (Rienecker et al., 2011), and the NCEP/NCAR reanalysis (Kalnay et al., 1996). Fujiwara et al. (2016) provides a review of key aspects of these data sets. Table 2 summarizes information on spatial and temporal resolution and coverage. The



four data sets considered here vary substantially in resolution, i. e., by a factor of 2 in temporal resolution, by a factor of 5 in vertical resolution, and by a factor of $20 \times 20$ in horizontal resolution. Note that we retrieved the data sets at the temporal and spatial resolution at which they are typically provided to the users by the respective centers. The same data sets have been considered by Hoffmann et al. (2016a), who provide a more detailed description of data preprocessing.

The Concordiasi balloon measurements covered the final stratospheric warming and decay of the polar vortex during 2010/2011 austral spring to summer. Although a mid-winter minor sudden stratospheric warming during July and early August 2010 resulted in an off-pole displacement and weakening of the stratospheric polar vortex (De Laat and van Weele, 2011; Klekociuk et al., 2011), the polar vortex returned to be relatively stable from mid-August to October, except for a second short warming that began in early September. This pattern was primarily attributed to the quasi-biennial oscillation being in a strong

positive phase that helped to maintain a persistent polar vortex. According to NASA Ozone Watch and the World Meteorological Organisation Antarctic Ozone Bulletins (see http://www.wmo.int/pages/prog/arep/gaw/ozone/index.html; last access: 30 September 2016), the longitudinally averaged poleward eddy heat flux between 45°S and 75°S, which is an indicator of disturbance in polar stratosphere, was much smaller than the long-term mean (Fig. 3), indicating that the vortex was relatively unperturbed from mid-September to December.

Figure 4 illustrates that the polar vortex was typically symmetric and stable in September and October. Afterwards, the polar vortex elongated and weakened gradually through November, was displaced off the pole in mid-December and broke down by mid-January 2011. The vortex breakup was marked when the winds around the vortex edge decreased below $15 \, \mathrm{m \, s^{-1}}$ on the 475 K potential temperature surface. From an analysis of temperatures on the levels where most of the balloon measurements were attained (about $50 - 60 \, \mathrm{hPa}$, $\sim 475 \, \mathrm{K}$), the final warming started from mid-October with development of strong zonal

asymmetries in temperature. The cold pool over the South Pole declined and displaced, and until end of November, minimum temperatures over Antarctica increased from around 180 to 220 K. A warm pool with temperatures of $230 - 240 \, \mathrm{K}$ dominated Antarctica from end of December. Consistent with the warming process, the polar jet showed a pronounced reduction in wind speed from $70 \, \mathrm{m \, s^{-1}}$ at the beginning of September to $40 \, \mathrm{m \, s^{-1}}$ in early December and then further weakened to less than $20 \, \mathrm{m \, s^{-1}}$ from beginning of January.

## 2.3   Trajectory calculations

We conducted the trajectory calculations for the Concordiasi balloon observations with the Lagrangian particle dispersion model MPTRAC (Hoffmann et al., 2016a). MPTRAC has been developed to support analyses of atmospheric transport processes in the free troposphere and stratosphere. In previous studies it was used to perform transport simulations for volcanic eruptions and to reconstruct time- and height-resolved emission rates for these events (Heng et al., 2016; Hoffmann et al.,

2016a). Transport is simulated by calculating trajectories for large numbers of air parcels based on given wind fields from global meteorological reanalyses. Turbulent diffusion and subgrid-scale wind fluctuations are simulated based on the Langevin equation, closely following the approach implemented in the Flexible Particle (FLEXPART) model (Stohl et al., 2005). Additional modules allow us to simulate sedimentation and the decay of particle mass, but they were not used here. The model is





particularly suited for large-scale simulations on supercomputers due to its efficient Message Passing Interface (MPI) / Open Multi-Processing (OpenMP) hybrid parallelization.

Trajectory calculations are based on numerical integration of the kinematic equation of motion,

$$\frac{d\mathbf{x}}{dt} = \mathbf{v}(\mathbf{x},t), \tag{1}$$

where $\mathbf{x}$ denotes the position and $\mathbf{v}$ the velocity of an air parcel at time $t$. The air parcel position $\mathbf{x}$ is defined by geographic latitude $\phi$ and longitude $\lambda$ as horizontal coordinates as well as pressure $p$ as vertical coordinate. The horizontal wind $(u,v)$ and vertical velocity $(\omega = dp/dt)$ at position $\mathbf{x}$ and time $t$ are obtained by linear spatial and temporal interpolation of the meteorological data. The kinematic equation of motion is solved with the explicit midpoint method,

$$\mathbf{x}(t+\Delta t) = \mathbf{x}(t) + \mathbf{v}\left(\mathbf{x}(t) + \frac{\Delta t}{2}\,\mathbf{v}\left(\mathbf{x}(t),t\right), t + \frac{\Delta t}{2}\right)\Delta t. \tag{2}$$

The time step $\Delta t$ mainly controls the trade-off between accuracy and speed of the calculations. For our simulations we selected $\Delta t = 30\,\text{s}$, which is sufficiently small so that truncation errors can be neglected. This time step is also consistent with the sampling rate of the balloon data.

The diffusion module of MPTRAC considers two processes. Turbulent diffusion is modelled by means of uncorrelated, Gaussian random displacements of the air parcels with zero mean and standard deviations $\sqrt{D_x \Delta t}$ and $\sqrt{D_z \Delta t}$, where $D_x$

and $D_z$ are the horizontal and vertical diffusion coefficients, respectively. Typical values for the stratosphere are $D_x = 0$ and $D_z = 0.1\,\text{m}^2\,\text{s}^{-1}$, according to choices made for the FLEXPART model (Legras et al., 2003; Stohl et al., 2005). Unresolved subgrid-scale wind fluctuations are most relevant for long-range simulations. These fluctuations are correlated over time and simulated with a Markov model, following the approach of Maryon (1998) and Stohl et al. (2005). For example, the zonal wind fluctuations $u'$ of each air parcel are calculated according to

$$u'(t+\Delta t) = r\,u'(t) + \sqrt{(1-r^2)\,\alpha\,\sigma_u^2}\,\xi, \tag{3}$$

with $r = \exp(-2\Delta t/\Delta t_{\text{met}})$ being a correlation coefficient depending on the model time step $\Delta t$ and the time interval $\Delta t_{\text{met}}$ of the meteorological data (3 or 6 h), $\alpha$ being a scaling factor used for downscaling of grid-scale variances $\sigma_u^2$ to subgrid scales, and $\xi$ being a Gaussian random variate with zero mean and unity variance. The FLEXPART model uses a default value of $\alpha = 0.16$ for downscaling of the grid-scale variances (or 40% in terms of standard deviations). Meridional wind and vertical velocity fluctuations are calculated in the same way.

For this study we implemented a new module in MPTRAC that allows us to simulate the vertical motions of the balloons more realistically. This module is called at each time step and adjusts the pressure of the air parcels so that vertical motions are constrained to either (i) an isobaric surface (constant pressure), (ii) an isopycnic surface (constant density), (iii) an isentropic surface (constant potential temperature), or (iv) the pressure time series measured by the balloon. In a first approximation the balloons move on isopycnic surfaces, which is represented by option (ii). However, the real dynamics of the balloons are more

complex, in particular if they encounter small-scale structures such as gravity waves (Vincent and Hertzog, 2014). On longer time scales it needs to be considered that there are diurnal variations in the balloon density as well as overall mass variations





due to the release of dropsondes (Sect. 2.1). These issues are partly circumvented by constraining the vertical motions to the balloon pressure data, which is represented by option (iv).

## 2.4 Validation approach

Although some of the Concordiasi balloon flights can be used to validate trajectory calculations for time periods as long as three months, we focused on shorter time windows. By splitting the balloon flights into smaller subsets of data, each containing 15 days of observations, we significantly increased the number of samples and improved the statistical accuracy of the results of the short-term validation. To further increase the number of samples we also allowed for overlap of the time windows, i. e., we shifted the 15-day windows in steps of 5 days. A shift of 5 days between the windows was selected, because trajectory errors are usually larger than the effective resolution of the meteorological data sets after that time. This means we can consider the results of overlapping windows as being statistically independent. We varied the starting days for the analysis of the different flights to homogenize temporal coverage. As there are data gaps in the GPS and TSEN data of the balloon measurements, we imposed the requirement that each sample should have at least 90% coverage. Based on these criteria we obtained a set of 104 samples of 15-day time windows from the 19 Concordiasi balloon flights.

Absolute horizontal transport deviations (AHTDs) and relative horizontal transport deviations (RHTDs) are standard measures to compare trajectory calculations with observations or to evaluate results for different model configurations (Kuo et al., 1985; Rolph and Draxler, 1990; Stohl et al., 1995; Stohl, 1998). While other measures of trajectory error have also been defined, AHTDs and RHTDs are most often reported because they can be compared easily to other studies. The AHTD at travel time $t$ of the trajectories is calculated as

$$\text{AHTD}(t) = \frac{1}{N_s \, N_e} \sum_{i=1}^{N_s} \sum_{j=1}^{N_e} \sqrt{\left[X_{i,j}(t) - x_i(t)\right]^2 + \left[Y_{i,j}(t) - y_i(t)\right]^2}, \qquad (4)$$

where $N_s$ refers to the number of reference trajectories and $N_e$ refers to the size of the ensemble of test trajectories that is to be evaluated for each reference trajectory. The coordinates $(X_{i,j}, Y_{i,j})$ and $(x_i, y_i)$ with $i = 1, \ldots, N_s$ and $j = 1, \ldots, N_e$ refer to the horizontal positions of the test and reference trajectories, respectively. Equation (4) is applied in different ways in this study. For instance, it is used to evaluate transport deviations between a single model trajectory and a balloon trajectory for just one sample ($N_s = 1$ and $N_e = 1$), between model and balloon trajectories for all samples ($N_s = 104$ and $N_e = 1$), or for dispersion simulations ($N_s = 104$ and $N_e = 1000$). Note that we calculated horizontal distances as Euclidean distances of the air parcel positions projected to the Earth's surface. RHTDs are calculated by dividing the AHTD of individual air parcels by the length of the corresponding reference trajectory. Absolute and relative vertical transport deviations (AVTDs and RVTDs) are defined similarly, based on pressure differences converted into vertical distances by means of the barometric formula.



## 3 Results

### 3.1 Direct validation of meteorological data

In this section we focus on the validation of temperatures and horizontal winds directly at the positions of the Concordiasi balloons. For this analysis the meteorological data are interpolated to the balloon positions by means of a 4-D linear interpo-

lation in space and time. This interpolation scheme is most commonly applied in state-of-the-art Lagrangian transport models (Bowman et al., 2013). Table 3 presents the summary statistics of meteorological data minus Concordiasi balloon observations. Note that we first discuss the results for low-pass filtered data, to exclude the effects of small-scale fluctuations due to gravity waves and turbulence (Sect. 2.1). These effects will be discussed separately at the end of this section. Our analysis shows that all data sets have a positive temperature bias, which is in the range of 0.4 K (ECMWF OA) to 2.1 K (NCEP/NCAR). Zonal

wind biases are in the range of $-0.3\,\mathrm{m\,s^{-1}}$ (NCEP/NCAR) to $0.5\,\mathrm{m\,s^{-1}}$ (MERRA). Meridional wind biases are about $0.1\,\mathrm{m\,s^{-1}}$ for all data sets. Standard deviations vary between 0.5 K (ECMWF OA) and 1.4 K (NCEP/NCAR) for temperature, $0.9\,\mathrm{m\,s^{-1}}$ (ECMWF OA) and $2.3\,\mathrm{m\,s^{-1}}$ (NCEP/NCAR) for the zonal wind, and $0.9\,\mathrm{m\,s^{-1}}$ (ECMWF OA) and $1.9\,\mathrm{m\,s^{-1}}$ (NCEP/NCAR) for the meridional wind. Skewness and excess kurtosis values found here indicate that the distributions are quite symmetric and not affected by many or rather large outliers. We confirmed these findings by calculating additionally more robust and resistant

statistical measures of location, spread, and symmetry, namely the median, the interquartile range, and the Yule-Kendall index (Wilks, 2011). These robust measures provided a quite similar picture to the standard measures.

Figure 5 shows bias and standard deviations of temperature and horizontal winds at different latitudes averaged over the entire time period of the campaign. All meteorological data sets show an increasing temperature bias from mid to high latitudes. The temperature warm bias at $80-85°\mathrm{S}$ is largest for NCEP/NCAR (3.1 K), followed by MERRA (1.4 K), ERA-Interim

(1.1 K), and ECMWF OA (0.5 K). At $60-65°\mathrm{S}$ the temperature biases range from 0.2 to 1.2 K. Temperature standard deviations do not show variation with latitude (ECMWF OA and ERA-Interim) or just a slight decrease from mid to high latitudes (MERRA and NCEP/NCAR). Zonal wind biases are below $\pm0.7\,\mathrm{m\,s^{-1}}$ and meridional wind biases are below $\pm0.3\,\mathrm{m\,s^{-1}}$ for all latitude bands considered here. Standard deviations of the zonal and meridional wind do not vary significantly with latitude for ECMWF OA and ERA-Interim. MERRA shows latitudinal variation in the range of $1.3-1.9\,\mathrm{m\,s^{-1}}$ for the zonal wind and

$1.2-1.6\,\mathrm{m\,s^{-1}}$ for the meridional wind. NCEP/NCAR mostly shows increasing standard deviations from high to mid latitudes, ranging from 1.8 to $3.0\,\mathrm{m\,s^{-1}}$ for the zonal wind and from 1.5 to $2.4\,\mathrm{m\,s^{-1}}$ for the meridional wind.

Figure 6 shows bias and standard deviations of temperature and horizontal winds for different months averaged over all latitudes. The temperature bias is at a maximum in September (1.3 K for ERA-Interim and 0.8 K for ECMWF OA) or October (2.7 K for NCAR/NCEP and 1.3 K for MERRA). Temperature standard deviations remain rather constant in the range from 0.5

to 0.9 K (ECMWF OA, ERA-Interim, and MERRA) or increase from 0.9 K in September to 1.6 K in December (NCEP/NCAR). Zonal wind biases vary more for MERRA and NCEP/NCAR, with a range of $\pm0.9\,\mathrm{m\,s^{-1}}$, and less for ECMWF OA and ERA-Interim, with a range of $\pm0.3\,\mathrm{m\,s^{-1}}$. Meridional wind biases remain below $\pm0.2\,\mathrm{m\,s^{-1}}$ for all months. The standard deviations of the zonal and meridional winds remain rather constant at about $0.8-1.0\,\mathrm{m\,s^{-1}}$ for ECMWF OA and ERA-Interim, but tend to increase from September to December for MERRA and NCEP/NCAR. In December we found maximum standard deviations of





$2.7\,\mathrm{m\,s^{-1}}$ (NCEP/NCAR) and $1.7\,\mathrm{m\,s^{-1}}$ (MERRA) for the zonal wind and $2.1\,\mathrm{m\,s^{-1}}$ (NCEP/NCAR) and $1.6\,\mathrm{m\,s^{-1}}$ (MERRA) for the meridional wind.

Overall, the direct validation of the large-scale features of the meteorological analyses in the Antarctic lower stratosphere provides satisfactory results. The summary statistics presented here are similar to those of earlier campaigns using superpres-
sure balloons observations, in particular with respect to results presented by Boccara et al. (2008) for the Vorcore campaign in 2005. Temperature biases of meteorological analyses at the southern hemisphere winter pole are well-known phenomena, which was reported also for other winters (Gobiet et al., 2005; Parrondo et al., 2007; Boccara et al., 2008). Using GPS radio occultation measurements in June to August 2003, Gobiet et al. (2005) showed that temperature biases of ECMWF analyses over the southern hemisphere winter pole vary with altitude. They found a warm bias of up to $3.5\,\mathrm{K}$ at $18-19\,\mathrm{km}$ (close to
the altitude of the Concordiasi balloon observations), a cold bias of up to $-3\,\mathrm{K}$ at $21-22\,\mathrm{km}$, and a warm bias of up to $3.5\,\mathrm{K}$ at $26-27\,\mathrm{km}$. Gobiet et al. (2005) speculate that the assimilation of microwave radiances from satellite measurements into the ECMWF analyses may be a reason for the temperature bias. We note that a warm bias is still present in Antarctic winter 2010 in all data sets, but its magnitude is significantly reduced for ECMWF OA, which may be attributed to improvements of the forecast model, data assimilation scheme, and observations used to produce this analysis. Although many factors in-
fluence the accuracy and precision of meteorological analyses, Table 3 indicates that the spatiotemporal resolution is a rather important factor. Both bias and standard deviations are lowest for ECMWF OA, which has the highest resolution, followed by ERA-Interim, MERRA, and finally NCEP/NCAR, which has the lowest resolution.

Finally, we also analyzed the effects of the low-pass filter that was applied to remove small-scale fluctuations from the data. Table 4 provides standard deviations of unfiltered minus filtered temperatures and horizontal winds. For comparison we also
provided standard deviations of unfiltered minus filtered balloon data in Table 4, which are considered as a measure of real small-scale fluctuations in the atmosphere. In fact, the balloon observations are an excellent source of data to study gravity waves (e. g., Hertzog et al., 2008, 2012; Plougonven et al., 2013; Vincent and Hertzog, 2014; Jewtoukoff et al., 2015). While large-scale biases are not affected by filtering and therefore not reported here, a comparison of standard deviations allows us to assess how well small-scale fluctuations are represented in the meteorological data sets. For the high resolution ECMWF OA
data the standard deviations removed by filtering are largest and about the same size as the standard deviations related to the differences of meteorological data minus balloon data. We found that ECMWF OA reproduces about 60% and ERA-Interim and MERRA about 30% of the standard deviations of the temperature and wind fluctuations of the balloons. NCEP/NCAR reproduces about 15% for temperature and 30% for the winds. This is in good agreement with the studies of Jewtoukoff et al. (2015), which found that ECMWF analyses underestimate gravity wave momentum fluxes derived from the Concordiasi
balloon observations by a factor of 5, and Hoffmann et al. (2016b), which found that wave amplitudes in the ECMWF analyses are typically underestimated by a factor of $2-3$ compared to Atmospheric InfraRed Sounder (AIRS/Aqua) observations.

## 3.2 Analysis of vertical motions

In the following sections of this paper we focus on the validation of trajectory calculations using the MPTRAC model with Concordiasi superpressure balloon observations. As outlined in Sect. 2.3, we implemented several new options in the MPTRAC



model to constrain the vertical motions of the air parcels. We first tried to identify the approach that is best suited to simulate the vertical motions of the balloons in a realistic manner. In our comparison we considered vertical motions based on prescribed pressure time series as measured by the balloons, isopycnic motions, isentropic motions, and vertical motions prescribed by the vertical velocities of the meteorological data sets (referred to as 'omega velocities' below). The comparison was conducted using ERA-Interim data as input for the trajectory calculations. For illustration, Fig. 7 shows examples of trajectories calculated with different types of vertical motions and the corresponding balloon observations. Within 15 days the balloon is advected by the polar night jet over a distance of nearly 30 000 km and encircles the south pole more than twice. At the end of the simulations we found horizontal transport deviations of about 30 km (0.1%) using the balloon pressure, 100 km (0.3%) for the isopycnic approach, 350 km (1.2%) for the isentropic approach, and 400 km (1.3%) for the omega velocity. In this particular example the balloon trajectory is reproduced with excellent accuracy by all simulations. We picked this example for presentation because the simulations are not strongly affected by any individual, complex meteorological conditions. In the example the balloon trajectory is best reproduced by constraining vertical movements based on the balloon pressure measurements or by using the isopycnic approach, as expected from the balloon dynamics (Sect. 2.1). Larger transport deviations are found using omega velocities and the isentropic approach. However, note that the trajectories based on omega velocities and the isentropic approach are in good agreement with each other, which was expected as atmospheric motions are isentropic on short time scales.

In order to take into account statistical variations, Fig. 8 shows transport deviations calculated from 104 samples of 15-day balloon trajectories of the Concordiasi campaign, which we selected according to the approach outlined in Sect. 2.4. The AHTDs increase rather steadily to about 1610 – 1750 km after 15 days. Like in the example shown in Fig. 7, the results cluster in two groups. Trajectories calculated using the balloon pressure and the isopycnic approach are similar to each other and yield results at the lower end of the AHTD ranges. Trajectories calculated using omega velocities and the isentropic approach are also similar to each other and yield results at the upper end of the AHTD ranges. The corresponding RHTDs are in a range of 4.1 – 5.2 % after 2 days and increase to 6.8 – 7.4% after 15 days. The mean difference between the two groups of simulations is about 0.7 percentage points. Note that RHTDs are quite large during the first 12 – 24 h, which is not representative, because the calculations are based on rather short reference trajectories. In addition, Fig. 8 also shows vertical transport deviations based on the isopycnic and isentropic approach as well as omega velocities. The AVTDs of the isopycnic approach increase steadily to about 200 m after 15 days. The corresponding RVTDs converge at 6 – 7% after 4 days. The AVTDs using omega velocities and the isentropic approach increase rapidly during the first 2 days and then increase more slowly up to 560 – 680 m after 15 days. The corresponding RVTDs converge to 17 – 21%. A possible reason for larger initial errors using omega velocities and the isentropic approach are uncertainties in the initial pressure values used to define the trajectory seeds. Simulations based on omega velocities or the isentropic approach are more strongly affected by short-term fluctuations of the initial pressure values than simulations based on the isopycnic approach. To mitigate uncertainties caused by short-term fluctuations, we used the mean pressure of the first 3 h of each balloon trajectory for initialization. However, our analysis indicates that vertical motions are best calculated using either the balloon pressure measurements or the isopycnic approach. For the remaining analyses we decided to calculate the trajectories using the balloon pressure measurements because these take into account any changes in the overall mass configuration of the balloon-gondola system (Sect. 2.1).





### 3.3 Impact of different meteorological analyses on trajectory calculations

In this section we present a comparison of transport deviations obtained with different meteorological data sets. Figures 9 and 10 show two examples of 15-day trajectory calculations using ECMWF OA, ERA-Interim, MERRA, and NCEP/NCAR data. The examples mainly serve to illustrate the large range of variability found in different simulations. For flight number 2

the simulated trajectories reproduce the observed balloon trajectory quite well. We found maximum AHTDs in the range of $650 - 1050\,\mathrm{km}$ and maximum RHTDs in the range of $3 - 7\%$ for the different data sets. Note that the maxima occur on different days, i. e., simulated trajectories may first deviate from and then approach the observed trajectories again. Despite being shorter (i. e., $12\,700\,\mathrm{km}$ versus $29\,700\,\mathrm{km}$), the simulated trajectories for flight number 12 deviate much larger from the observations. Here we found maximum AHTDs of $3100 - 5200\,\mathrm{km}$ and maximum RHTDs of $53 - 70\%$. The two examples illustrate the large

variability between different samples, which is attributed to situation-dependent factors, including the individual meteorological conditions. A large number of independent samples needs to be analyzed in order to obtain statistically significant results.

Figure 11 shows transport deviations for the different meteorological data sets calculated from 104 samples of 15-day trajectories (Sect. 2.4). In contrast to the individual examples, we found that the AHTDs increase rather steadily over time, which suggests that outliers play a minor role and that the statistics are robust. After 15 days the AHTDs are close to $2200\,\mathrm{km}$

for NCEP/NCAR, $1800\,\mathrm{km}$ for MERRA, $1600\,\mathrm{km}$ for ERA-Interim, and $1400\,\mathrm{km}$ for ECMWF OA. From Fig. 11 we can also estimate the growth rates of the AHTDs. The growth rate for NCEP/NCAR is close to $170\,\mathrm{km\,day^{-1}}$ for the first 12 days, but slightly decreases thereafter. For MERRA we found a growth rate of $120\,\mathrm{km\,day^{-1}}$. The growth rates of both ERA-Interim and ECMWF OA are close to $60\,\mathrm{km\,day^{-1}}$ during the first 5 days, close to $110\,\mathrm{km\,day^{-1}}$ during day $6 - 12$, and get more variable during day $13 - 15$. The RHTDs of NCEP/NCAR decrease from about $12\%$ after day 1 to about $8.5\%$ after day 15.

The RHTDs of MERRA decrease from $10\%$ to $7.5\%$. The RHTDs of both ECMWF OA and ERA-Interim slightly increase from $4.5\%$ to $6 - 7\%$. These results agree well with those reported by Boccara et al. (2008) for the Vorcore campaign in 2005. For 15 days' trajectory time Boccara et al. (2008) found mean spherical distances of about $1650\,\mathrm{km}$ (with an interquartile range of $800 - 3600\,\mathrm{km}$) for ECMWF analyses (sampled at $0.5° \times 0.5°$ horizontal resolution and 60 levels vertically) and $2350\,\mathrm{km}$ ($1400 - 3800\,\mathrm{km}$) for NCEP/NCAR data. The transport deviations and growth rates found here also compare well with a

wider range of results for the troposphere reported by Stohl (1998). Our analysis indicates that the best accuracy of trajectory calculations in the Antarctic lower stratosphere is achieved with ECMWF OA and ERA-Interim, followed by MERRA and NCEP/NCAR. This is related to the accuracy of the horizontal winds of the meteorological data sets as discussed in Sect. 3.1.

### 3.4 Impact of subgrid-scale wind fluctuations

In this section we discuss the influence of diffusion on the trajectory calculations. We assessed this by means of dispersion

simulations, each consisting of 1000 trajectories, and by applying the MPTRAC diffusion module described in Sect. 2.3. Note that these simulations consider only horizontal diffusion, because vertical motions have been restricted to the pressure measurements of the balloons. Following Stohl et al. (2005), the turbulent horizontal diffusivity coefficient in the stratosphere was set to zero, $D_x = 0$, i. e., the diffusion in our simulations is related only to horizontal subgrid-scale wind fluctuations. For



comparison with diffusion-free simulations, two examples of dispersion simulations are also shown in Fig. 9. For flight number 2 we found only minor spread of the air parcels due to diffusion whereas for flight number 12 it is quite substantial, illustrating that diffusion may vary significantly from case to case. The examples also suggest that the uncertainties of the trajectory calculations are linked to the meteorological situation, as low diffusion goes along with good accuracy of the trajectories for
flight number 2 whereas high diffusion goes along with low accuracy for flight number 12.

Kahl (1996) analyzed correlations between trajectory model errors and the complexity of the meteorological situation under study in more detail. He quantified the complexity of the meteorological conditions by means of the so-called 'meteorological complexity factor' (MCF), which measures the dispersion of a set of stochastic trajectories generated by random perturbations superimposed upon an observed wind field. Kahl (1996) pointed out that trajectory errors are representative only if they are
larger than the corresponding MCF. Similar to Kahl (1996), we estimated the MCF of our simulations by applying Eq. (4) to the trajectory ensemble. However, instead of taking the balloon trajectory as a reference, the MCF was calculated using a simulated trajectory without diffusion as a reference. The simulated reference trajectory is usually close to the ensemble mean because the deviations of the ensemble trajectories are often symmetric around the ensemble mean. The MCFs of the four meteorological data sets of our study are shown in Fig. 11. The MCFs increase rather steadily over time. After 15 days we
found values of 1300 km for ECMWF OA, 800 – 900 km for MERRA and NCEP/NCAR, and 600 km for ERA-Interim. These differences in the MCFs came somewhat unexpected, as the spread of air parcels ideally should be the same in all simulations, independent of the meteorological data set and the diffusion model being applied. The differences are not directly related to the resolution of the meteorological data sets, as can be seen from the ranking of the MCFs of the data sets. The inconsistencies of the MCFs found here might be due to dynamical inconsistencies of the analysis wind fields that are introduced during the data
assimilation process. Such dynamical inconsistencies may lead to more rapid dispersion and spurious mixing in Lagrangian transport model simulations (Stohl et al., 2004).

In principle, we may tune the scaling factor $\alpha$ in Eq. (3) of the MPTRAC diffusion module to achieve simulations with more consistent MCFs. However, we refrained from any tuning measures, because appropriate reference data for validation are lacking. We applied a constant scaling factor $\alpha = 0.16$ in all simulations, which is the default value used in the FLEXPART model.
However, despite the different levels of MCFs found in the simulations, we conclude that the transport deviations between the simulations and the balloons are representative, because the are generally larger than the MCFs. To further confirm this result we also calculated the AHTDs between the trajectory ensembles and the balloon trajectories. We found that the transport deviations with or without diffusion are rather similar (Fig. 11). The AHTDs for ERA-Interim, MERRA, and NCEP/NCAR differ less than ±50 km. For ECMWF OA the AHTDs with diffusion are up to 200 km larger than the AHTDs without diffusion.
We attribute this to the fact that simulated diffusion is largest for ECMWF OA, as indicated by the corresponding MCFs. This shows that diffusion does not induce any significant uncertainties in our analysis of transport deviations. The results remain meaningful, even if diffusion is not explicitly taken into account.



## 4 Summary and conclusions

In this study we validated temperature and horizontal wind data of the ECMWF operational analysis (OA) and the ERA-Interim, MERRA, and NCEP/NCAR reanalyses at southern hemisphere mid and high latitudes (about $60 - 85°$S) in the lower stratosphere (about $17 - 18.5$ km). The validation was based on Concordiasi superpressure balloon observations in September 2010 to January 2011. We found temperature warm biases of the analyses in the range of $0.4 - 2.1$ K, which are similar to the values found in earlier studies (Gobiet et al., 2005; Parrondo et al., 2007; Boccara et al., 2008). Zonal and meridional wind biases are below $\pm 0.5$ m s$^{-1}$. After applying a low-pass filter to remove small-scale fluctuations due to gravity waves and turbulence, standard deviations of analyses minus observations are in the range of $0.4 - 1.4$ K for temperature and $0.9 - 2.3$ m s$^{-1}$ for the winds. Overall, these are satisfactory validation results that are comparable to other studies using superpressure balloon observations in the Antarctic lower stratosphere (e. g., Boccara et al., 2008). Note that ECMWF OA, ERA-Interim, and MERRA validation results for Antarctica are much better than those found by Podglajen et al. (2014) for the equatorial lower stratosphere. As Podglajen et al. (2014) and this study both used observations gathered in 2010, this provides further evidence that the quality of meteorological analyses tends to degrade from high latitudes towards the Equator. Podglajen et al. (2014) showed that the lower quality of the reanalyses at low latitudes is associated with poor representation of large-scale equatorial waves, which might be improved by more direct observations of stratospheric wind profiles over wide regions along the equatorial belt. Considering four different meteorological data sets in our study, we found clear indications that spatial and temporal resolution of the data and truncation of the models also play an important role in determining accuracy and precision of the analyses. Best large-scale accuracy and precision are achieved by ECMWF OA (highest resolution), followed by ERA-Interim, MERRA, and NCEP/NCAR (lowest resolution). Model truncation also affects the representation of small-scale fluctuations. Standard deviations of unfiltered minus filtered temperature and wind data of the balloons are reproduced at a level of about 60% by ECMWF OA, but only by $15 - 30$% by the reanalyses. For ECMWF OA temperatures this is consistent with recent studies of Jewtoukoff et al. (2015) and Hoffmann et al. (2016b), providing further evidence that the ECMWF operational model explicitly resolves a significant portion of the atmospheric gravity wave spectrum.

We also used the Concordiasi balloon observations to validate trajectory calculations in the Antarctic lower stratosphere. Some difficulties are related to the fact that the overall mass configuration of the balloon-gondola system may change during the flight. Our analysis showed that balloon trajectories are best reproduced by the isopycnic approach or by nudging vertical motions to the pressure measurements of the balloons. In this study we analyzed 104 samples of trajectories from 19 balloon flights for time periods of 15 days. Relative horizontal transport deviations are in the range of $4.5 - 7$% for ECMWF OA and ERA-Interim, $7.5 - 10$% for MERRA, and $8.5 - 12$% for NCEP/NCAR. Growth rates of absolute horizontal transport deviations are in the range of $60 - 110$ km day$^{-1}$ for ECMWF OA and ERA-Interim, about $120$ km day$^{-1}$ for MERRA, and about $170$ km day$^{-1}$ for NCEP/NCAR. These results agree well with those of Boccara et al. (2008) for the Vorcore campaign in 2005. They show a significant improvement compared to early studies of Knudsen and Carver (1994) and Knudsen et al. (1996), which found transport deviations of about 20% between trajectories based on ECMWF analyses and long-duration balloon observations. We conducted the trajectory calculations with the Lagrangian particle dispersion model MPTRAC (Hoffmann et al.,





2016a); and our study provides a contribution to the validation of this new model. However, the results will be transferable also to other Lagrangian transport models for the stratosphere. We used the diffusion module of MPTRAC to conduct dispersion simulations. The analysis revealed some difficulties with the modelling approach for subgrid-scale wind fluctuations and the wind data driving the simulations, as the spread of air parcel trajectories simulated with different meteorological data sets was

not consistent. Future work may comprise additional analyses and may focus on tuning of the subgrid-scale parametrization scheme. Selected examples of dispersion simulations indicate that the accuracy of trajectory calculations is linked to meteorological complexity, as suggested by Kahl (1996). In this study we analyzed a rather large number of trajectory samples, though, and the effects of meteorological complexity averaged out and did not alter the results of the analysis of transport deviations significantly. Future studies applying chemistry-transport models to assess the dynamics of the polar vortex or to investigate

polar ozone loss may use our validation results as additional guideline for error analysis.

## 5   Code and data availability

The quality-controlled meteorological TSEN data set is available from Laboratoire de Météorologie Dynamique (LMD) from their web site at http://www.lmd.polytechnique.fr/VORCORE/McMurdoE.htm (last access: 21 December 2016). The ERA-Interim reanalysis and operational analyses are distributed by the European Centre for Medium-Range Weather Forecasts

(ECMWF), see http://www.ecmwf.int/en/forecasts/datasets (last access: 21 December 2016). MERRA data are provided by the Global Modeling and Assimilation Office at NASA Goddard Space Flight Center through the NASA GES DISC online archive, see http://disc.sci.gsfc.nasa.gov/mdisc/data-holdings/merra (last access: 21 December 2016). NCEP/NCAR reanalysis data were obtained from the NOAA/OAR/ESRL PSD, Boulder, Colorado, USA, from their web site at http://www.esrl.noaa.gov/psd (last access: 21 December 2016). The code of the Massive-Parallel Trajectory Calculations (MPTRAC) model is available

under the terms and conditions of the GNU General Public License, Version 3 from the repository at https://github.com/slcs-jsc/mptrac (last access: 21 December 2016).

*Author contributions.*   All authors contributed to the design of the study and provided input to the manuscript. LH conducted the transport simulations and the scientific analysis. AH provided support regarding the scientific analysis of the Concordiasi superpressure balloon observations. TR and OS were responsible for preprocessing of the meteorological data. XW provided the characterization of the meteorological

conditions during the campaign.

*Acknowledgements.*   Concordiasi was built by an international scientific group and is currently supported by the following agencies: Météo-France, CNES, IPEV, PNRA, CNRS/INSU, NSF, NCAR, Concordia consortium, University of Wyoming, and Purdue University. ECMWF also contributes to the project through computer resources and support, and scientific expertise. The two operational polar agencies PNRA and IPEV are thanked for their support at Concordia station. Concordiasi is part of the THORPEX-IPY cluster within the International Polar

Year effort. The authors acknowledge the Jülich Supercomputing Centre (JSC) for providing computing time on the supercomputer JURECA.





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



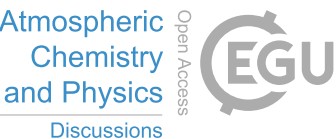

**Table 1.** Concordiasi Balloon Flights over Antarctica in September 2010 to January 2011

| Flight Number | Flight Code | Gondola ID | Flight Start | Flight End |
|---|---|---|---|---|
| 1 | MSD01 | 10V01N46 | 2010/09/23 | 2010/12/11 |
| 2 | MSD02 | 10V02N48 | 2010/09/23 | 2010/11/18 |
| 3 | MSD03 | 10V03N39 | 2010/10/15 | 2010/11/04 |
| 4 | MSD04 | 10V04N40 | 2010/09/24 | 2010/12/27 |
| 5 | MSD05 | 10V05N44 | 2010/09/25 | 2010/12/22 |
| 6 | MSD06 | 10V06N37 | 2010/09/28 | 2010/12/09 |
| 7 | MSD07 | 10V07N41 | 2010/09/30 | 2010/12/09 |
| 8 | MSD08 | 10V08N49 | 2010/10/26 | 2011/01/19 |
| 9 | MSD09 | 10V09N22 | 2010/10/07 | 2011/01/04 |
| 10 | MSD10 | 10V10N25 | 2010/10/14 | 2010/12/24 |
| 11 | MSD11 | 10V11N56 | 2010/10/19 | 2010/12/29 |
| 12 | MSD12 | 10V12N66 | 2010/10/20 | 2011/01/23 |
| 13 | MSD13 | 10V13N65 | 2010/10/19 | 2010/11/30 |
| 14 | PSC14 | 10V14N42 | 2010/09/15 | 2010/12/21 |
| 15 | PSC15 | 10V15N32 | 2010/09/08 | 2010/09/16 |
| 16 | PSC16 | 10V16N35 | 2010/09/11 | 2010/10/11 |
| 17 | PSC17 | 10V17N31 | 2010/09/14 | 2010/12/10 |
| 18 | PSC18 | 10V18N43 | 2010/09/29 | 2010/12/16 |
| 19 | PSC19 | 10V19N27 | 2010/10/08 | 2010/12/24 |





**Table 2.** Temporal and Spatial Resolution of Meteorological Data Sets

| Data Product | Temporal Resolution | Top Level | Vertical Levels | Horizontal Resolution |
|---|---|---|---|---|
| ECMWF OA | 3 h | 0.01 hPa | 91 | $0.125° \times 0.125°$ |
| ERA-Interim | 6 h | 0.1 hPa | 60 | $1.000° \times 1.000°$ |
| MERRA | 3 h | 0.1 hPa | 42 | $1.250° \times 1.250°$ |
| NCEP/NCAR | 6 h | 10 hPa | 17 | $2.500° \times 2.500°$ |





**Table 3.** Statistics of Meteorological Analyses Minus Concordiasi Balloon Observations (Based on $N \approx 2.52 \times 10^6$ Measurements)

|  | ECMWF OA | ERA-Interim | MERRA | NCEP/NCAR |
|---|---|---|---|---|
| Temperature [K] |  |  |  |  |
|     Bias | 0.4 | 0.8 | 1.1 | 2.1 |
|     Standard Deviation | 0.5 | 0.6 | 0.9 | 1.4 |
|     Skewness | 0.5 | 0.4 | -0.2 | -0.8 |
|     Excess Kurtosis | 2.2 | 1.0 | -0.1 | 1.0 |
| Zonal Wind [m s$^{-1}$] |  |  |  |  |
|     Bias | 0.1 | 0.3 | 0.5 | -0.3 |
|     Standard Deviation | 0.9 | 1.0 | 1.6 | 2.3 |
|     Skewness | 0.1 | 0.1 | -0.4 | -0.4 |
|     Excess Kurtosis | 1.4 | 1.5 | 2.3 | 1.5 |
| Meridional Wind [m s$^{-1}$] |  |  |  |  |
|     Bias | 0.1 | 0.1 | 0.1 | 0.1 |
|     Standard Deviation | 0.9 | 0.9 | 1.4 | 1.9 |
|     Skewness | 0.0 | 0.0 | -0.3 | 0.0 |
|     Excess Kurtosis | 1.8 | 3.3 | 2.9 | 1.5 |

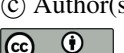



**Table 4.** Standard Deviations of Unfiltered Minus Filtered Meteorological Data

|  | Balloons | ECMWF OA | ERA-Interim | MERRA | NCEP/NCAR |
|---|---|---|---|---|---|
| Temperature [K] | 0.7 | 0.4 | 0.2 | 0.2 | 0.1 |
| Zonal Wind [m s$^{-1}$] | 1.5 | 0.9 | 0.4 | 0.4 | 0.4 |
| Meridional Wind [m s$^{-1}$] | 1.6 | 1.0 | 0.5 | 0.5 | 0.5 |





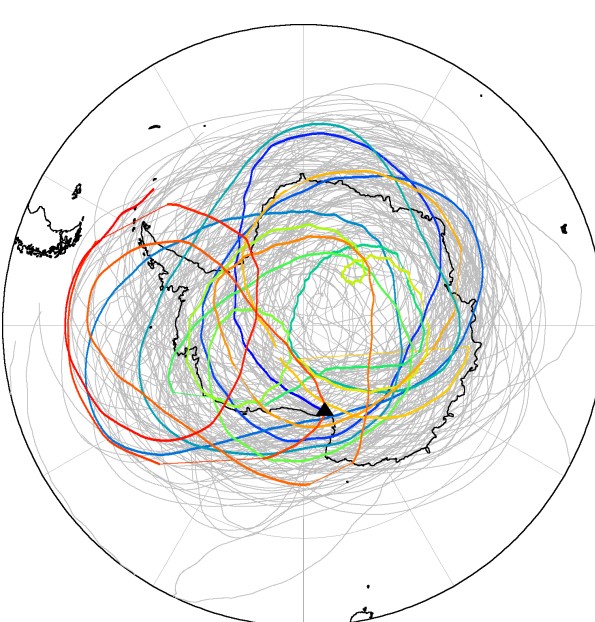

**Figure 1.** Map of superpressure balloon trajectories (gray curves) during the Concordiasi field campaign in Antarctica in September 2010 to January 2011. The colored curve highlights the trajectory of flight number 4, with colors from blue to red indicating measurement time. The black triangle shows the location of McMurdo station.





**Figure 2.** Time series of meteorological data of flight number 4 of the Concordiasi campaign (see Fig. 1). Grey curves show unfiltered data from GPS and TSEN measurements at 30 s time intervals. Black curves show results of a low-pass filter with 15 h cut-off frequency. Inset plots show data for 14 – 17 November 2010.





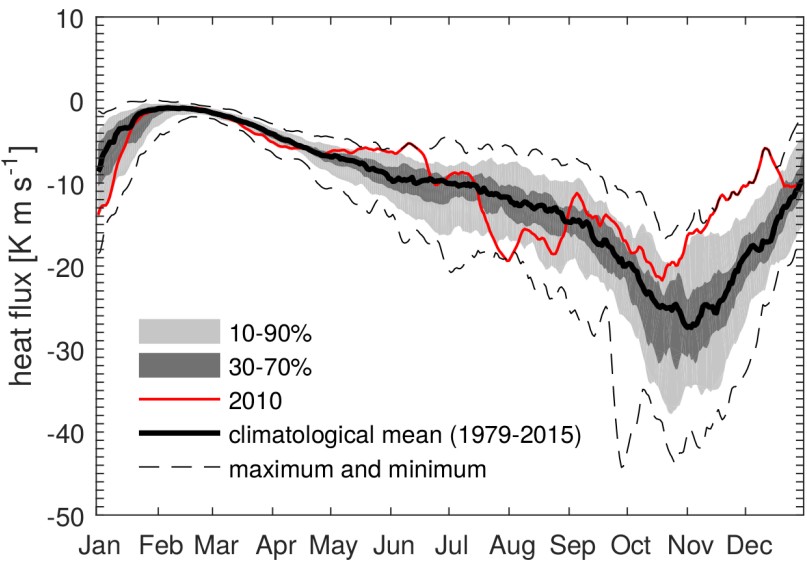

**Figure 3.** Activity of the polar vortex at 50 hPa as represented by the 45-day running mean of the eddy heat flux between 45 and 75°S. The red curve shows results for the year 2010. Black and gray curves illustrate statistics of the long-term mean (1979 – 2015). Data were obtained from NASA Ozone Watch from their web site at https://ozonewatch.gsfc.nasa.gov (last access: 16 December 2016).







**Figure 4.** ERA-Interim potential vorticity ($1\,\mathrm{PVU} = 10^{-6}\,\mathrm{K\,m^2\,s^{-1}\,kg^{-1}}$; shaded) and zonal wind contours ($\mathrm{m\,s^{-1}}$; black curves) on the 475 K isentropic surface. Data are shown for 0 UTC on selected days.





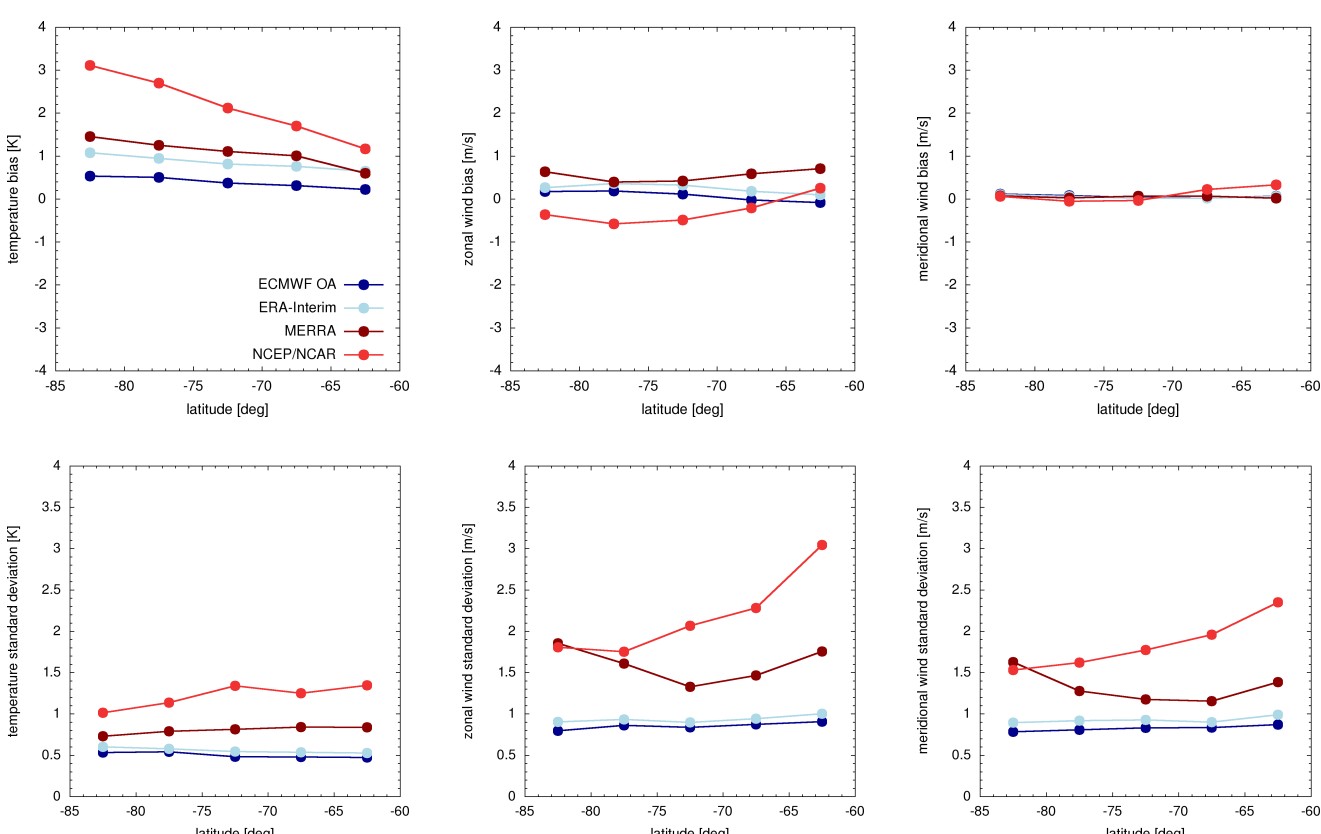

**Figure 5.** Bias and standard deviations of temperature and horizontal winds of meteorological analyses minus Concordiasi balloon data for different latitudes.





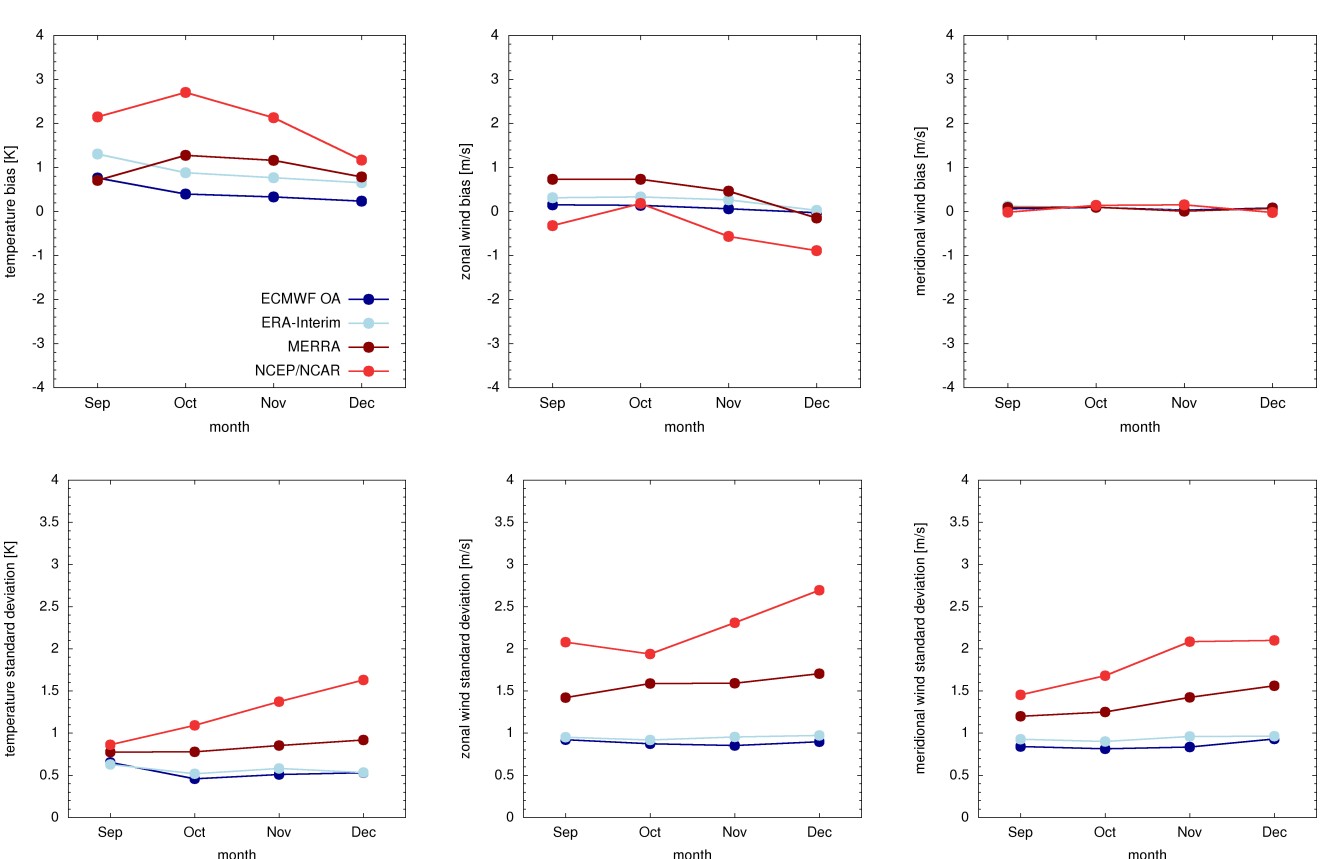

**Figure 6.** Same as Fig. 5, but for different months.





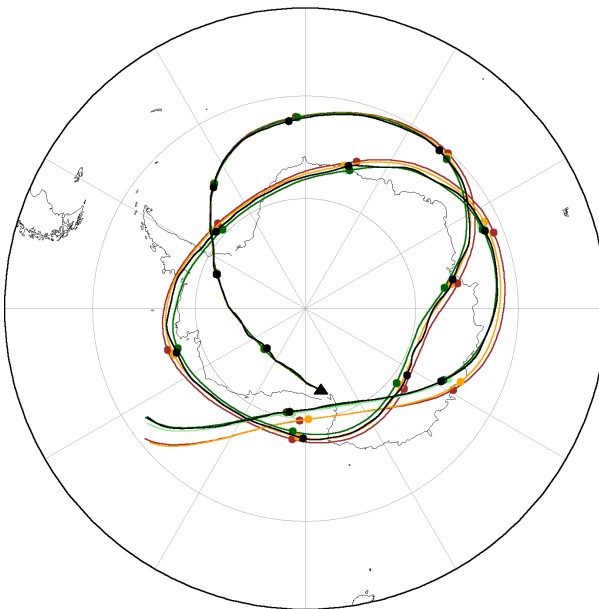

**Figure 7.** Comparison of 15-day trajectories calculated with different types of vertical motion (dark green: balloon pressure, light green: isopycnic, orange: isentropic, red: omega velocity) and corresponding Concordiasi balloon trajectory (black). The plot title provides the starting time and the triangle indicates the starting position of the trajectories. Circles indicate trajectory positions at 0 UTC each day.





**Figure 8.** Transport deviations of simulated and observed balloon trajectories for different types of vertical motion. Trajectories were calculated with ERA-Interim horizontal winds. The analysis is based on 104 samples of 15-day trajectories from the Concordiasi campaign.





**Figure 9.** Examples of trajectories calculated with different meteorological analyses (dark blue: ECMWF OA, light blue: ERA-Interim, dark red: MERRA, light red: NCEP/NCAR) and corresponding Concordiasi balloon trajectory (black). Plot titles provide the starting times and triangles indicate the starting positions of the trajectories. Circles indicate trajectory positions at 0 UTC each day. Plots at the top show individual trajectories calculated without diffusion. Plots at the bottom illustrate dispersion simulations with diffusion being considered.





**Figure 10.** Absolute (top) and relative (bottom) horizontal transport deviations of the trajectories shown in Fig. 9 (top).







**Figure 11.** Horizontal transport deviations of simulated and observed balloon trajectories for different meteorological analyses (top). Dotted gray lines represent AHTD growth rates of 60, 120, and 170 km day$^{-1}$. Also shown are the meteorological complexity factor for dispersion simulations (bottom, left) and the AHTD differences due to diffusion (bottom, right).