# Peer review of "Validation of meteorological analyses and trajectories in the Antarctic lower stratosphere using Concordiasi superpressure balloon observations"

_Atmospheric Chemistry and Physics, 2017_

## Referee Comment (RC1) · A. Stohl (Referee) · 17 Feb 2017

This is an interesting study that compares meteorological data from several (re-)analysis data sets to balloon measurements in the stratosphere. While the results are perhaps not entirely unexpected and the methods applied are not particularly novel, the study is solid and it is always good to see validation of re-analysis data (and trajectory calculations based on them) against independent data. The paper would, however, benefit from some shortening of the Results section (see also comments below). Most of the results are relatively easy to interpret and do not need to be discussed in such

great detail as there is now – the figures are to a large extent self-explanatory, given that the analysis methods and statistical quantities presented are relatively simple. I have some more detailed comments below but if they are addressed adequately, I am in favor of publication of this paper.

Major points to consider:

Please confirm explicitly (both in your response and in the paper) that the Concordiasi balloon data was not assimilated in any of the data sets that you are using. I assume this is the case but if the data were assimilated, of course your study would not be very meaningful as the data could not be considered independent.

Section 3.1: What is the motivation for comparing the (re)analysis data sets against smoothed balloon data rather than against the unfiltered data? This reduces the relative differences between the high-resolution operational analyes and the coarser resolution re-analysis data because the operational data captures some of the high-resolution variability, while the coarser-resolution re-analyses capture very little of it. Thus, by filtering the performance of the higher-resolution data sets is artificially degraded (relative to the other data sets), and that doesn't seem to be very logical. I think it would be much more meaningful to do the standard/main/reference comparisons against the unfiltered data and use the filtering as a sensitivity study to show that the poorer performance of the coarser-resolution data sets is due to their inability to capture some of the fine-resolution details.

The results section is in many ways too detailed. For example, in section 3.1 (but also in other sections) there are too many numbers that the reader can't all remember. These numbers are all available in Figures and if summaries are needed, this information could be put in tables. However, putting so many numbers into the text, makes it very exhausting to read. I would suggest to substantially shorten this by concentrating on the main findings/messages and the conclusions that can be drawn from these results, rather than listing all individual results. References to the figures should be enough.

Section 3.2: Here, it is stated that the best results are found when constraining the vertical position of the trajectories to the actual pressure heights of the balloons. But this should not be presented as a finding, as it is a trivial result. Anything else than that would indicate some error in the calculation! Notice also that this method has been used before and this may be mentioned. I think the first publication of this was by Baumann and Stohl (1997) but there have been other uses, e.g. Riddle et al. (2006). Related to this, it is also a trivial result that the AVTDs of the isopycnic approach increase with time (page 10, line 25). This does not indicate any real errors, but just shows the fact that the trajectory height is not constrained to the balloon altitude, and since the model does not (cannot) account for altitude variations of the balloon, of course there are errors.

Section 3.4: It would be interesting to calculate the AHTDs also for the ensemble-mean trajectory of all the trajectories with superimposed subgrid-scale fluctuations. Are the AHTDs for this trajectory larger than for the reference trajectory without sub-grid wind fluctuations?

Minor points:

Introduction, first few lines: When speaking of the polar vortex, make clear you mean the southern hemisphere.

Page 3, line 33: you say that GPS positions are recorded at each observation time. I suppose this means every 30 seconds, but where you write this, it is not really clear, as you mention the other observations only later. So explicitly say how often GPS data are recorded.

Figures 5 and 6: The ranges used for the y axes are not ideal. This is extreme for the upper right panels (meridional wind bias) where the data range occupies clearly less than 10% of the available space. This makes it very difficult to actually read the values and makes the figure almost meaningless.

СЗ

Is figure 10 really needed?

Typos, etc.:

Page 4, line 23: evelopE

Page 12, line 26: ... because theY are ...

References:

Baumann, K., and A. Stohl (1997): Validation of a long-range trajectory model using gas balloon tracks from the Gordon Bennett Cup 95. J. Appl. Meteor. 36, 711-720.

Riddle, E. E., P. B. Voss, A. Stohl, D. Holcomb, D. Maczka, K. Washburn, and R. W. Talbot (2006): Trajectory model validation using newly developed altitudecontrolled balloons during the International Consortium for Atmospheric Research on Transport and Transformations 2004 campaign. J. Geophys. Res. 111, D23S57, doi:10.1029/2006JD007456.

---

## Referee Comment (RC2) · Anonymous Referee #1 · 22 Feb 2017

Review:

Summary: This study provides validation of four lower stratospheric temperature and wind analyses through comparisons with independent (not incorporated into the analyses) long duration constant pressure balloon observations from Sep 2010 to Dec 2011. In addition to direct temperature and wind comparisons, the study compares trajectory errors, a quantity that is important to understand for the interpretation of stratospheric transport and chemistry. The observations, analyses, and methods are clearly presented and the results clearly explained. Past work is appropriately referenced, figures

are clear, and the paper is very well written. The balloons provide an excellent source of independent, in situ, stratosphere observations, that are well exploited by this study. Overall this work should be of interest to many readers of ACP.

Main Point:

MERRA-2 was released in the Fall of 2015 and should really be included in place of, or better yet, in addition to the MERRA results. MERRA-2 replaces MERRA and will be used in more future studies than the previous, and no longer produced, MERRA system. Including MERRA-2 comparisons should make the paper much more useful and more widely cited. It is difficult to recommend publications as is, with only the out-of-date MERRA system included.

Minor Points:

Table 2: The MERRA and MERRA-2 products are also provided on the 72 model vertical levels. The higher vertical resolution available may change the balloon comparison results. Was there a reason for examining the pressure level output?

Figure 4, Caption: Add text for the latitude of the outer circle and the longitude orientation.

Line 23, "40 m/s": There are still some 50 m/s contours in the 2010-12-01 figure. Is the 40 m/s number an average?

Line 21, "grid-scale variances". How are the grid-scale variances calculated? Are they space or time variances?

Figures 5 and 6: The meridional wind bias plot might show differences more clearly with a different vertical scale. The values are small and the circulation will change over the balloon record and with latitude, however the small average meridional wind error is still of interest.

Figures in general: The multi panel figures should be labelled a, b, c, ... and referred

to as such in the figure captions and text.

---

## Referee Comment (RC3) · Anonymous Referee #3 · 20 Mar 2017

The manuscript is generally well written with nice figures and a clear presentation of the methods applied. But unfortunately I consider the applied method to be flawed. By having access to the operational ECMWF analysis feedback data and to the ECMWF ERA-Interim analysis feedback archive, I can confirm that the Concordiasi temperature and wind observations were assimilated by both data assimilation systems. ECMWF assimilated the data distributed on the GTS (15 minutes frequency). 68% of the data was assimilated and 32% removed by thinning.

The ERA-Interim statistics for the whole Concordiasi campaign showed balloon data

minus analysis departures for temperature: Standard deviation 0.7K, bias -0.3K. Compared against 12-hour background fields the values were: Standard deviation 0.8K, bias -0.5K. The similar statistics for zonal wind: analysis departures: Standard Deviation 1.2 m/s, bias -0.1 m/s. Background departures: Standard deviation 1.9 m/s, bias -0.1 m/s. Meridional wind: analysis departures: standard deviation 1.2 m/s, bias 0 m/s. Background departures: Standard deviation: 1.9 m/s, bias -0.1 m/s. Around 41000 temperature Concordiasi measurements were assimilated during the three months. 41500 zonal and 41500 Concordiasi meridional measurements were assimilated.

These detailed statistics are included here to confirm that the Concordiasi data was fitted well by the ERA-Interim analysis and therefore cannot be considered independent data. Similarly can be said for the operational ECMWF analysis (not shown). This means that this is not a valid comparison of the four (re)analysis systems, if it is true the data was not assimilated in NCEP reanalysis and the MERIS reanalysis. This means that the core part of the manuscript, the inter-comparison, would not make much sense and would not be fair. Based on this I would recommend that the editor rejects the paper.

Additional costly assimilation experiments without assimilation of the Concordiasi in the ECMWF systems would be required for a fair comparison. It would require a very significant rewrite of the manuscript to remove all the parts that relates to inter-comparison, or clearly split the description and evaluation of the ECMWF systems' results and NCEP/MERIS. No matter what it would not provide a proper inter-comparison.

At this stage it does not make sense for me to provide detailed comments. The two main issues I have are related to use of interpolated model data and the 15 hour time filtering.

---

## Referee Comment (RC4) · A. Stohl (Referee) · 20 Mar 2017

This comment is to support the suggestion of reviewer #3 that the manuscript needs to be rejected, unfortunately.

I already wrote in my original review: "Please confirm explicitly (both in your response and in the paper) that the Concordiasi balloon data was not assimilated in any of the data sets that you are using. I assume this is the case but if the data were assimilated, of course your study would not be very meaningful as the data could not be considered independent."

Printer-friendly version

[Figure]

As reviewer #3 reports in his review, Concordiasi data were in fact assimilated by the ECMWF data assimilation system. Thus, unless the authors can prove the opposite, I think there is no other choice than to reject the paper. It is of course a pity as the study is otherwise well designed, but the authors should have checked this very critical point more carefully.

———————————————————

---

## Short Comment (SC1) · 27 Mar 2017

The comment was uploaded in the form of a supplement:
http://www.atmos-chem-phys-discuss.net/acp-2017-71/acp-2017-71-SC1-supplement.pdf

---

## Editor Comment (EC1) · F. Khosrawi (Editor) · 18 Apr 2017

Dear authors and referees,

I would like to thank the referees for their thorough review and the authors for their clarifications. The fact that Concordiasi balloon data has been assimilated into the meteorological analyses cannot be ignored. However, in my opinion this does not justify a rejection. I am quite confident that the study can be brought into a publishable form with major revisions.

[Figure]

The authors presented already some ideas how they could improve their manuscript. Most important is that the fact that the Concordiasi balloon data is assimilated in the meteorological analyses is considered when performing the assessment and drawing conclusions. Nevertheless, the Concordiasi data are not the only data that is assimilated and one should not forget that meteorological analyses are based on model simulations. Thus, even with the Concordiasi data assimilated into the analyses the impact cannot be that severe that one cannot do a meaningful assessment of the performance of the meteorological data sets.

Contrary to the suggestion by the authors to remove the NCAR/NCEP I would suggest to keep this data set to have one "independent" data set in the comparison. To make the assessment then more concise the section could be split into two comparisons: one between NCEP and Concordiasi and another one comparing the Concordiasi data with ECMWF OP, ERA-Interim, MERRA and MERRA-2. A second option would be to just compare the meteorological analyses without comparing these to the Concordiasi balloon data. Further, I would appreciate if MERRA would not just be replaced with MERRA-2, but rather that both data sets would be used in the assessment. Another third option would be to include another independent data set into the comparison.

Another point that could be improved is the references to previous studies. There are a lot of studies comparing the performance of meteorological analyses by Gloria Manney and her colleagues (see list below). Additionally, I would suggest to change the term validation in evaluation or assessment throughout the manuscript.

Based on the suggestions for improvements given by the referees, by myself and by the authors themselves I would like to encourage the authors to conduct major revisions and resubmit their manuscript.

[Figure]

Best regards

Farahnaz Khosrawi

References:
Lawrence et al. (2015), Comparisons of polar processing diagnostics from 34 years of the ERA-Interim and MERRA reanalyses, Atmospheric Chemistry and Physics, Vol. 15, Issue 7, 38723-3892.

Manney et al. (2005): Diagnostic comparison of meteorological analyses during the 2002 Antarctic winter, Monthly Weather Review, Vol. 133. Issue 5, 1261-1278.

Manney et al. (2003): Lower stratospheric temperature differences between meteorological analyes in two cold Arctic winters and their impact on polar processing studies, Journal of Geophysical Research, Vol. 108, Issue 5.

Manney et al. (1996): Comparison of U. K. Meteorological Office and U. S. National Center stratospheric analyses during northern and southern winter, Journal of Geophysical Research, Vol. 101, Issue D6, 10311-10334.

---

## Author Comment (AC1) · 18 Apr 2017

Dear editor and reviewers,

following the first review comment of Andreas Stohl, we contacted scientists and support staff at ECMWF, NASA, and NCEP/NCAR to clarify the question if the Concordiasi data have been assimilated by the respective centres.

Unfortunately and unexpectedly, it turned out that this was case for both ECMWF data sets and MERRA, but not for NCEP/NCAR. In particular, we learned that 15-min time averaged data from the Concordiasi balloons have been transmitted over the Global

Telecommunication System (GTS) (Rabier, 2013). The data transmitted over GTS were then assimilated by the respective centers.

We agree that the Concordiasi data can not be considered as an independent data source for direct validation of the large-scale state of the ECMWF products and MERRA.

However, we think that after clarifying the role of the data assimilation in this case, the data can still be used for a useful assessment and intercomparison of the meteorological data sets along these lines:

1) Restrict the intercomparison to ECMWF products and MERRA, but remove NCEP/NCAR from the list. Indeed it would not be a fair comparison, if the Concordiasi data have been assimilated for ECMWF and MERRA, but not for NCEP/NCAR. Following a comment of reviewer #2, we already started to analyze MERRA-2 data, which we would like to include in this assessment instead.

2) Remove statements referring to studies of comparisons with other balloon campaigns, such as Vorcore and PreConcordiasi, where data have not been considered for data assimilation, for the same reason.

3) Analyses are a result of using various observations (satellite and in-situ), a forecast model, and an assimilation procedure. A comparison with the assimilated balloon data does not provide validation in a strict sense, but it still provides information regarding the performance of the overall system. Our study showed notable and significant differences between the ECMWF operational analysis, ERA-Interim, and MERRA, despite the fact that the balloon data have been assimilated (with a potentially large impact). Observing system experiments could help to clarify this question, but are beyond our capabilities. Therefore we propose to significantly shorten the discussion regarding the direct comparison of the analyses and the balloon data in Sect. 3.

4) Instead, we would like to put the focus of the paper on the assessment of representation of small-scale structures in the analyses due to gravity waves, a topic which is already addressed in Sect. 3 of the paper. This analysis remains valid despite the fact that the Concordiasi data have been assimilated, because they were subject to downsampling and/or data thinning. At ECMWF about 40,000 observations have been assimilated according to reviewer #3, whereas our assessment of small-scale structures considered the nearly complete data set of about 2,500,000 observations.

5) Clarify in Sect. 4 that another major aim of the study is the validation of trajectory calculations with the rather new Lagrangian particle dispersion model MPTRAC itself, which was the main reason of conducting this study initially. This purpose can be fulfilled with the Concordiasi GPS balloon tracks, despite the fact that wind and temperature measurements have been assimilated.

We think that a major revision of the paper as outlined above is possible in the remaining time frame. However, before we conduct this revision, we would like to ask for editor approval.

Best regards

Lars Hoffmann (on behalf of all authors)

Reference

Rabier, Florence, et al. (2013), The Concordiasi field experiment over Antarctica: first results from innovative atmospheric measurements., B. Am. Meteo. Soc., 94.3, ES17-ES20, doi:10.1175/BAMS-D-12-00005.1.
* * *

---

## Author Response (AR1)

**Replies to review comments**

*We thank the reviewers, the co-editor, as well as Cameron Homeyer and his students for the thoughtful comments and the time and effort spent on the manuscript. Please find our point-by-point replies below (in blue color and italics). A revised manuscript with tracked changes was attached.*

*In this initial reply we would like to address the main issue regarding the assimilation of the Concordiasi data into the (re)analyses.*

*Following the first review comment by Andreas Stohl, we contacted scientists and support staff at ECMWF, NASA, and NCEP/NCAR to clarify the question if the Concordiasi data have been assimilated by the respective centres.*

*Unfortunately and unexpectedly, it turned out that this was case for the ECMWF data sets and MERRA, but not for NCEP/NCAR. In particular, we learned that 15-min time averaged data from the Concordiasi balloons have been transmitted over the Global Telecommunication System (GTS) (Rabier, 2013). The data transmitted over GTS were then assimilated by the respective centers.*

*We agree that the Concordiasi data can not be considered as an independent data source for direct validation of the large-scale state of the ECMWF products and MERRA. They can only be used for validation of NCEP/NCAR. Furthermore, we consider the data useful for the assessment of small-scale structure (e. g., gravity waves), because the Concordiasi data have been subject to downsampling and data thinning before they were assimilated.*

*Meteorological analyses are a result of combining various observations (satellite and in-situ), a forecast model, and a data assimilation procedure. A comparison with the assimilated balloon data does not provide validation in a strict sense, but it still provides information regarding the performance of the overall system. Our study showed notable differences between the ECMWF operational analysis, ERA-Interim, and MERRA, despite the fact that the balloon data have been assimilated.*

*After seeking consent with the co-editor, we therefore conducted a major revision of the manuscript. In particular, we made the following changes:*

1. *We added a new paragraph in Sect. 2.2 describing the role of data assimilation in this study.*

2. *We replaced the term "validation" by "comparison" or "evaluation" throughout the manuscript.*

3. *We significantly shortened Sect. 3.1 of the paper that dealt with the "direct validation" of the meteorological analysis.*

4. *We removed some of the comparisons to other balloon campaigns (Vorcore and Pre-Concordiasi), where data have not been considered for data assimilation.*

**Reviewer #1**

Summary: This study provides validation of four lower stratospheric temperature and wind analyses through comparisons with independent (not incorporated into the analyses) long duration constant pressure balloon observations from Sep 2010 to Dec 2011. In addition to direct temperature and wind comparisons, the study compares trajectory errors, a quantity that is important to understand for the interpretation of stratospheric transport and chemistry. The observations, analyses, and methods are clearly presented and the results clearly explained. Past work is appropriately referenced, figures are clear, and the paper is very well written. The balloons provide an excellent source of independent, in situ, stratosphere observations, that are well exploited by this study. Overall this work should be of interest to many readers of ACP.

Main Point:

MERRA-2 was released in the Fall of 2015 and should really be included in place of, or better yet, in addition to the MERRA results. MERRA-2 replaces MERRA and will be used in more future studies than the previous, and no longer produced, MERRA system. Including MERRA-2 comparisons should make the paper much more useful and more widely cited. It is difficult to recommend publications as is, with only the out-of-date MERRA system included.

*MERRA-2 data were quite new when we conducted the study, but we included them in this revision.*

Minor Points:

Table 2: The MERRA and MERRA-2 products are also provided on the 72 model vertical levels. The higher vertical resolution available may change the balloon comparison results. Was there a reason for examining the pressure level output?

*Following Hoffmann et al. (2016), we initially focused on MERRA output on pressure levels, because the MPTRAC model uses pressure as vertical coordinate. However, for the assessment of MERRA-2, we implemented new code to process meteorological data on hybrid sigma/pressure levels.*

Figure 4, Caption: Add text for the latitude of the outer circle and the longitude orientation.

*We added corresponding text in the caption.*

Line 23, "40 m/s": There are still some 50 m/s contours in the 2010-12-01 figure. Is the 40 m/s number an average?

*This was an error in the text. We replaced "early December" by "mid of December".*

Line 21, "grid-scale variances". How are the grid-scale variances calculated? Are they space or time variances?

*For each particle position the grid-scale variance was calculated considering values of the 8 nearest grid points and 2 nearest time steps of the meteorological data. We rephrased the text to clarify.*

Figures 5 and 6: The meridional wind bias plot might show differences more clearly with a different vertical scale. The values are small and the circulation will change over the balloon record and with latitude, however the small average meridional wind error is still of interest.

*We adjusted the vertical scale to make small values visible.*

Figures in general: The multi panel figures should be labelled a, b, c, ... and referred to as such in the figure captions and text.

*These labels will be inserted during the copy-editing process as needed.*

**Reviewer #2 (A. Stohl)**

This is an interesting study that compares meteorological data from several (re)analysis data sets to balloon measurements in the stratosphere. While the results are perhaps not entirely unexpected and the methods applied are not particularly novel, the study is solid and it is always good to see validation of re-analysis data (and trajectory calculations based on them) against independent data. The paper would, however, benefit from some shortening of the Results section (see also comments below). Most of the results are relatively easy to interpret and do not need to be discussed in such great detail as there is now – the figures are to a large extent self-explanatory, given that the analysis methods and statistical quantities presented are relatively simple. I have some more detailed comments below but if they are addressed adequately, I am in favor of publication of this paper.

Major points to consider:

Please confirm explicitly (both in your response and in the paper) that the Concordiasi balloon data was not assimilated in any of the data sets that you are using. I assume this is the case but if the data were assimilated, of course your study would not be very meaningful as the data could not be considered independent.

*Please see initial reply regarding the role of data assimilation.*

Section 3.1: What is the motivation for comparing the (re)analysis data sets against smoothed balloon data rather than against the unfiltered data? This reduces the relative differences between the high-resolution operational analyses and the coarser resolution re-analysis data because the operational data captures some of the high resolution variability, while the coarser-resolution re-analyses capture very little of it. Thus, by filtering the performance of the higher-resolution data sets is artificially degraded (relative to the other data sets), and that doesn't seem to be very logical. I think it would be much more

meaningful to do the standard/main/reference comparisons against the unfiltered data and use the filtering as a sensitivity study to show that the poorer performance of the coarser-resolution data sets is due to their inability to capture some of the fine-resolution details.

*The filter was applied to achieve a consistent separation between large-scale dynamics (e. g., zonal temperature gradients and planetary waves) and small-scale features (mainly gravity waves). The cut-off period of 15 h was selected to cover the longest possible periods of inertial gravity waves at high latitudes. Please note that the application of a low-pass filter for detrending is a standard technique for gravity wave analyses. We think that such a scale separation is useful, because we may expect that large-scale features are significantly affected by data assimilation of the 15 min-downsampled balloon data, whereas for the small-scale features this is less relevant. We rephrased the text to clarify.*

The results section is in many ways too detailed. For example, in section 3.1 (but also in other sections) there are too many numbers that the reader can't all remember. These numbers are all available in Figures and if summaries are needed, this information could be put in tables. However, putting so many numbers into the text, makes it very exhausting to read. I would suggest to substantially shorten this by concentrating on the main findings/messages and the conclusions that can be drawn from these results, rather than listing all individual results. References to the figures should be enough.

*Section 3.1 was significantly shortened as many of the results were related to comparisons of large-scale dynamics of the reanalysis and the balloon data. Those results have been shortened or removed, because the data are not independent in most cases. Instead, we focus on the assessment of representation of small-scale structures in the analyses due to gravity waves.*

Section 3.2: Here, it is stated that the best results are found when constraining the vertical position of the trajectories to the actual pressure heights of the balloons. But this should not be presented as a finding, as it is a trivial result. Anything else than that would indicate some error in the calculation! Notice also that this method has been used before and this may be mentioned. I think the first publication of this was by Baumann and Stohl (1997) but there have been other uses, e.g. Riddle et al. (2006). Related to this, it is also a trivial result that the AVTDs of the isopycnic approach increase with time (page 10, line 25). This does not indicate any real errors, but just shows the fact that the trajectory height is not constrained to the balloon altitude, and since the model does not (cannot) account for altitude variations of the balloon, of course there are errors.

*Earlier studies using stratospheric superpressure balloon observations (e. g. Hertzog et al., 2004; Boccara et al., 2008) used only the isopycnic approach for trajectory evaluation. Our study shows for the first time that the isopycnic approach provides very similar results compared to trajectories constrained by pressure observations for this type of balloon. The studies of Baumann and Stohl (1997) and Riddle et al. (2006) refer to other types of altitude-controlled balloons, which are operating in the lower troposphere. However, we*

*rephrased the text in Sect. 3.2 and added the references to clarify.*

Section 3.4: It would be interesting to calculate the AHTDs also for the ensemble-mean trajectory of all the trajectories with superimposed subgrid-scale fluctuations. Are the AHTDs for this trajectory larger than for the reference trajectory without sub-grid wind fluctuations?

*The AHTDs for both cases are very similar, but not necessarily always larger if we consider superimposed subgrid-scale fluctuations. Because the AHTDs are so similar, we decided to show a plot of the differences in Fig. 10.*

Minor points:

Introduction, first few lines: When speaking of the polar vortex, make clear you mean the southern hemisphere.

*We clarified this in the revised manuscript.*

Page 3, line 33: you say that GPS positions are recorded at each observation time. I suppose this means every 30 seconds, but where you write this, it is not really clear, as you mention the other observations only later. So explicitly say how often GPS data are recorded.

*The GPS positions were recorded every 60 s and interpolated to 30 s time intervals to combine them with the Tsen data. We added this information in the text.*

Figures 5 and 6: The ranges used for the y axes are not ideal. This is extreme for the upper right panels (meridional wind bias) where the data range occupies clearly less than 10% of the available space. This makes it very difficult to actually read the values and makes the figure almost meaningless.

*We adjusted the y axes ranges to make small values visible.*

Is figure 10 really needed?

*Considering the information already given in the text, Figure 10 was somewhat redundant. We removed it from the revised manuscript.*

Typos, etc.:

Page 4, line 23: evelopE

*Fixed.*

Page 12, line 26: ... because theY are ...

*Fixed.*

**Reviewer #3**

The manuscript is generally well written with nice figures and a clear presentation of the methods applied. But unfortunately I consider the applied method to be flawed. By having access to the operational ECMWF analysis feedback data and to the ECMWF ERA-Interim analysis feedback archive, I can confirm that the Concordiasi temperature and wind observations were assimilated by both data assimilation systems. ECMWF assimilated the data distributed on the GTS (15 minutes frequency). 68% of the data was assimilated and 32% removed by thinning.

The ERA-Interim statistics for the whole Concordiasi campaign showed balloon data minus analysis departures for temperature: Standard deviation 0.7K, bias -0.3K. Compared against 12-hour background fields the values were: Standard deviation 0.8K, bias -0.5K. The similar statistics for zonal wind: analysis departures: Standard Deviation 1.2 m/s, bias -0.1 m/s. Background departures: Standard deviation 1.9 m/s, bias -0.1 m/s. Meridional wind: analysis departures: standard deviation 1.2 m/s, bias 0 m/s. Background departures: Standard deviation: 1.9 m/s, bias -0.1 m/s. Around 41000 temperature Concordiasi measurements were assimilated during the three months. 41500 zonal and 41500 Concordiasi meridional measurements were assimilated.

These detailed statistics are included here to confirm that the Concordiasi data was fitted well by the ERA-Interim analysis and therefore cannot be considered independent data. Similarly can be said for the operational ECMWF analysis (not shown). This means that this is not a valid comparison of the four (re)analysis systems, if it is true the data was not assimilated in NCEP reanalysis and the MERIS reanalysis. This means that the core part of the manuscript, the inter-comparison, would not make much sense and would not be fair. Based on this I would recommend that the editor rejects the paper.

Additional costly assimilation experiments without assimilation of the Concordiasi in the ECMWF systems would be required for a fair comparison. It would require a very significant rewrite of the manuscript to remove all the parts that relates to inter-comparison, or clearly split the description and evaluation of the ECMWF systems results and NCEP/MERIS. No matter what it would not provide a proper inter-comparison.

At this stage it does not make sense for me to provide detailed comments. The two main issues I have are related to use of interpolated model data and the 15 hour time filtering.

*Please see initial reply regarding the role of data assimilation.*

**Short comment by Cameron Homeyer et al.**

Disclaimer: This is a summary of a group peer review exercise in my senior undergraduate research class at the University of Oklahoma. 39 students participated in this review.

The authors present a validation and transport analysis of multiple large-scale models (both operational and reanalysis) using a (hopefully independent) set of long-duration stratospheric balloon observations over the Antarctic. Errors in stratospheric temperatures and winds are examined along the path of many balloon flights and found to be dependent on latitude and model grid resolution. Through the use of trajectory calculations with balloon locations as initial particle conditions, the authors identify errors in transport calculations using the model wind fields through comparisons with the observed balloon paths. Horizontal displacement errors in the trajectory calculations are found to scale considerably with grid resolution. Furthermore, multiple trajectory calculations for differing sources of vertical motion are calculated and show that horizontal and vertical displacement errors of trajectories also depend significantly on this choice.

Part of this study is a demonstration that the authors' recently developed trajectory model MPTRAC produces reasonable results, while the main focus is on comparisons between observed balloon flight paths and trajectory calculations driven by wind fields from models with varying complexity and grid resolution. One of the strengths of this manuscript is the quality of the figures included. We find that while the paper is generally well written, there are some areas of the technical description and analysis that are unclear or too vague. In some cases, this casts doubt on the results. Detailed comments are provided below.

General Comments

1. On the calibration of the balloon temperature sensor and its accuracy: It is mentioned in Section 2.1 that due to daytime heating by the sun, data from the thermistors on the balloon undergo an empirical correction. However, no detail on exactly how the data are corrected and how this impacts the uncertainty of the measurement is given. Precision of the temperature data set is given, but it seems that understanding its uncertainty and the effect of the empirical correction on the model validation carried out here are required.

*For solar zenith angles $\alpha \leq 94.5°$ (daytime measurements) the corrected temperature $T_c$ is deduced from the raw measurement $T$ through*

$$T_c = T - A \exp\left(\frac{\alpha - 94.5°}{B}\right),$$

*with empirical coefficients $A$ and $B$ for each thermistor. Nighttime measurements ($\alpha > 94.5°$) are not corrected, i.e., $T_c = T$. For further details we would refer to the detailed*

*description provided by Hertzog et al. (2004), which is cited in our manuscript. Note that we conducted a cross check by calculating day- and nighttime statistics separately, but did not find any significant differences (not shown in the paper). This suggests that the empirical temperature correction does not introduce any large uncertainties.*

2. Euclidean distances are used to determine horizontal displacement errors in the trajectories, but is this an appropriate choice? Since the curvature of zonal and meridional winds is most pronounced at the pole, shouldn't distances be calculated using a geodesic approach? Using a Euclidean approach may introduce unwanted errors that bias the results.

*The approach used here approximates spherical distances with $\geq 99\%$ accuracy for distances up to $3000\,km$ (Rößler et al., 2017). Therefore, no significant biases were introduced in our results.*

3. There is a substantial amount of unnecessary detail in the abstract, much of which (including lists of numerical values) seems better left to the main sections and tables of the paper.

*We tried to shorten the abstract in the revised manuscript, but we feel that numerical values should still be presented, because they summarize most of the rather detailed statistics presented in the paper.*

4. While the polar vortex was used to motivate this work, it would be nice to see some connection between the findings of this study and the polar vortex in Section 4. For example, how might the results from this work be leveraged in future studies examining dynamics and transport in the vortex?

*Following a suggestion of the co-editor, we revised Sect. 4 in order to better relate this work to studies of Lawrence et al. (2015) and Manney et al. (1996, 2003, 2005), which deal with the evaluation of reanalyses in the polar stratosphere.*

5. Is the Concordiasi dataset independent of those used for assimilation in the set of models analyzed here? This is an important point that was not discussed in the manuscript.

*Please see initial reply regarding the role of data assimilation.*

Specific Comments

Page 4, line 23: "envelop" should be "envelope"

*Fixed.*

Page 5, line 10: For clarity, it would be good to point out that the QBO positive phase is westerly here.

*We added this information.*

Page 8, lines 20-21: "deviations do not" should be "deviations either do not"

*Fixed.*

Page 11, line 18: Change "get" to "become"

*Fixed.*

Page 12, line 26: "because the are generally" should be "because they are generally"

*Fixed.*

Page 13, line 21: Change "but only by 15" to "but only of 15"

*Fixed.*

Figure 1: The objective of the colored balloon path is appreciated, but a scale should be given so the reader knows how changes in color correspond to changes in time.

*We added a color scale.*

Figure 3: While this figure is not a leading element of the analysis, it would be good to provide more detail on the dataset this is based on than including a link in the caption.

*We added the information that this analysis is based on MERRA-2.*

Figure 4: While the figure caption states the dataset used for this sequence of maps is ERA-Interim, it would be good to specify this in the corresponding text.

*As this figure is not dealing with an intercomparison of the different meteorological data sets, we mention the name of the specific data set only in its caption.*

**Co-editor comment (F. Khosrawi)**

Dear authors and referees,

I would like to thank the referees for their thorough review and the authors for their clarifications. The fact that Concordiasi balloon data has been assimilated into the meteorological analyses cannot be ignored. However, in my opinion this does not justify a rejection. I am quite confident that the study can be brought into a publishable form with major revisions.

The authors presented already some ideas how they could improve their manuscript. Most important is that the fact that the Concordiasi balloon data is assimilated in the meteorological analyses is considered when performing the assessment and drawing conclusions. Nevertheless, the Concordiasi data are not the only data that is assimilated and one should not forget that meteorological analyses are based on model simulations. Thus, even with the Concordiasi data assimilated into the analyses the impact cannot be that severe that one cannot do a meaningful assessment of the performance of the meteorological data sets.

*Please see initial reply regarding the role of data assimilation.*

Contrary to the suggestion by the authors to remove the NCAR/NCEP I would suggest to keep this data set to have one "independent" data set in the comparison. To make the assessment then more concise the section could be split into two comparisons: one between NCEP and Concordiasi and another one comparing the Concordiasi data with ECMWF OP, ERA-Interim, MERRA and MERRA-2. A second option would be to just compare the meteorological analyses without comparing these to the Concordiasi balloon data. Further, I would appreciate if MERRA would not just be replaced with MERRA-2, but rather that both data sets would be used in the assessment. Another third option would be to include another independent data set into the comparison.

*We followed the advise and kept NCEP/NCAR data in the paper and added MERRA-2 in addition to MERRA. We decided to not split the analysis in two parts (independent versus dependent data), but tried to make very clear in the discussion which parts are affected by data assimilation.*

Another point that could be improved is the references to previous studies. There are a lot of studies comparing the performance of meteorological analyses by Gloria Manney and her colleagues (see list below). Additionally, I would suggest to change the term validation in evaluation or assessment throughout the manuscript.

*We tried to improve the discussion in Sect. 4 by relating our work to these references. We also rephrased the term "validation" as suggested.*

Based on the suggestions for improvements given by the referees, by myself and by the authors themselves I would like to encourage the authors to conduct major revisions and resubmit their manuscript.

Best regards Farahnaz Khosrawi

[revised manuscript text omitted]

---

## Author Response (AR2)

**Replies to review comments**

*We thank the reviewer and the co-editor for the time and effort spent on the manuscript. Please find our point-by-point replies below (in blue color and italics). A revised manuscript with tracked changes was attached.*

**Reviewer #1**

The present version of the manuscript acknowledges the assimilation of the balloon data into the forecast systems. The scientific significance of the manuscript is thus substantially reduced compared to the original submission but the paper still serves at least the purpose of trajectory evaluation. The presentation quality has been improved and the limitations due to the data assimilation are clearly outlined. Thus, in summary and after reading the editor's arguments, I find the paper acceptable for ACP after two small changes:

Page 1, line 13: replace "different" simply with "the" (twice). The impact is not due to "different" models (or assimilation schemes) but simply due to the fact that models and assimilation schemes are used. Even with the same models and schemes, there would be differences to the observations. Same applies to page 8, line 30 – although there the phrasing is slightly better.

*Fixed on page 1 and page 8 as suggested.*

Page 5, line 22: before written twice

*Fixed.*

[revised manuscript text omitted]

**Figure 8.** Examples of trajectories calculated with different meteorological analyses (dark blue: ECMWF OA, light blue: ERA-Interim, dark red: MERRA-2, light red: MERRA, orange: NCEP/NCAR) and corresponding Concordiasi balloon trajectory (black). Plot titles provide the starting times and triangles indicate the starting positions of the trajectories. Circles indicate trajectory positions at 0 UTC each day. Plots at the top show individual trajectories calculated without diffusion. Plots at the bottom illustrate dispersion simulations with diffusion being considered.

[Figure]

**Figure 9.** Horizontal transport deviations of simulated and observed balloon trajectories for different meteorological analyses (top). Dotted gray lines represent AHTD growth rates of 60 and 170 km day$^{-1}$. Also shown are the meteorological complexity factor for dispersion simulations (bottom, left) and the AHTD differences that are introduced by adding diffusion (bottom, right).